# Towards Explainable Harmful Meme Detection through Multimodal Debate between Large Language Models

## ABSTRACT

The age of social media is flooded with Internet memes, necessitating a clear grasp and effective identification of harmful ones. This task presents a significant challenge due to the implicit meaning embedded in memes, which is not explicitly conveyed through the surface text and image. However, existing harmful meme detection methods do not present readable explanations that unveil such implicit meaning to support their detection decisions. In this paper, we propose an explainable approach to detect harmful memes, achieved through reasoning over conflicting rationales from both harmless and harmful positions. Specifically, inspired by the powerful capacity of Large Language Models (LLMs) on text generation and reasoning, we first elicit multimodal debate between LLMs to generate the explanations derived from the contradictory arguments. Then we propose to fine-tune a small language model as the debate judge for harmfulness inference, to facilitate multimodal fusion between the harmfulness rationales and the intrinsic multimodal information within memes. In this way, our model is empowered to perform dialectical reasoning over intricate and implicit harm-indicative patterns, utilizing multimodal explanations originating from both harmless and harmful arguments. Extensive experiments on three public meme datasets demonstrate that our harmful meme detection approach achieves much better performance than state-of-the-art methods and exhibits a superior capacity for explaining the meme harmfulness of the model predictions.

## KEYWORDS

harmful meme detection, explainability, multimodal debate, LLMs

## 1 INTRODUCTION

The increasing prevalence of social media has led to the emergence of a novel multimodal entity: *meme*. A meme consists of a picture combined or embedded with a concise textual component. Due to their ease of dissemination, memes have the capability to rapidly proliferate across various online media platforms. While memes are often humorously perceived, they become a potential source of harm when the amalgamation of the image and text is strategically employed to foster political and socio-cultural divisions.

Harmful memes[1] are generally defined as "multimodal units consisting of an image and accompanying text that has the potential to cause harm to an individual, an organization, a community, or the whole society" [48]. For instance, during the COVID-19 pandemic, a widely circulated meme shown in Figure 1 was produced by anti-vaccination groups via spoofing the image of Bill Gates. The widespread circulation of such multimodal scaremongering content about COVID-19 vaccines inflicted significant damage on both

---

[1]**Disclaimer:** This paper contains content that may be disturbing to some readers.

*Conference WWW '24, May 13–17, 2024, Singapore*
2023. ACM ISBN 978-x-xxxx-xxxx-x/YY/MM...$15.00
https://doi.org/10.1145/nnnnnnn.nnnnnnn

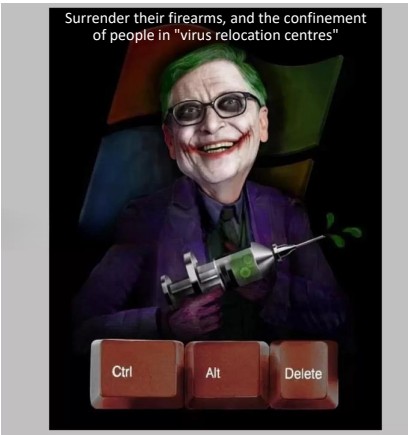

**Figure 1: Example of trending memes on social media. Meme text: *Surrender their firearms, and the confinement of people in "virus relocation centres".***

Bill Gates' personal reputation and the efforts to establish strong immune defenses[2]. Therefore, it becomes imperative to develop automatic approaches for harmful meme detection to effectively unveil the dark side of memes on the Web. This task, as suggested in [40], extends beyond mere analysis of meme images and texts in isolation. It demands a comprehensive examination, aiming not only to decipher their intrinsic semantics but also to provide the explainability of prediction results from the detection models.

Previous studies [22, 40] straightforwardly utilized pre-trained vision-language models [28, 34] to classify harmful meme by training additional task-specific classification layers. Pramanick et al. [41] proposed a multimodal framework to achieve state-of-the-art performance on harmful meme detection by modeling the deep multimodal interactions from the global and local perspectives. More recently, Cao et al. [5] proposed a prompt-based method with the meme text and image caption as the prompt for masked language modeling [10, 33] to predict whether the meme is harmful. A follow-up study [4] tailored additional hand-crafted questions as the prompt of frozen pre-trained vision-language models, to further improve image captioning for better meme classification performance. However, these approaches for harmful meme detection only capture superficial harmfulness patterns for classification in a black-box manner [15], which often overlooks or oversimplifies the supportive basis to explain the final harmfulness prediction.

Generally, understanding and analyzing memes poses a significant challenge due to their implicit meaning that is not explicitly conveyed through the surface text and image. Providing explanations for why a particular meme is deemed harmful, is crucial to the content moderation process on social media, as both moderators and users may want to comprehend the harmful content

---

[2]https://www.bbc.com/news/55101238

behind a flagged meme [21]. Nevertheless, a comprehensive explanation requires a deep understanding of commonsense and cultural context. For example, to explain the harmfulness of the meme in Figure 1, a human checker needs the socio-cultural knowledge that the character with a vaccine gun represents Bill Gates from Microsoft, who is often the target of anti-vaccination campaigners' memes due to his promotion of vaccine development; and also should know that the "Ctrl Alt Delete" key combination makes reference to the mandatory reboot function in Microsoft Windows, satirizing vaccine injection when combating the virus. In contrast, conventional detection models lack such natural sentences with multimodal reasoning chains, hindering their ability to provide informative explanations for harmfulness predictions.

We contend that the challenge lies in delivering clear and accurate explanations that consistently assist in deciphering the concealed semantics within the multimodal nature of memes. In this paper, we consider the following key principles in the design of our approach: 1) To capture implicit meanings of memes, we elicit and harness the rich prior knowledge embedded in Large Language Models (LLMs) [3, 6, 51]; 2) As the knowledge elicited directly from LLMs may exhibit variation and bias, we resort to a core element of human problem-solving, *i.e.*, debate, to stimulate dialectical thinking [2] among LLMs, thereby facilitating complex reasoning for enhancing the accuracy and explainability of harmful meme detection; 3) The semantic interaction between the meme and the harmfulness rationales extracted from the LLM debate could serve to augment multimodal feature representation, thereby fostering a deeper contextual understanding of the model in the context of harmfulness inference. To all these ends, we propose an **Explain**able approach for **H**armful **M**eme detection, ExplainHM, by leveraging the powerful text generation capacity of LLMs via Chain-of-Thought (CoT) prompting [24, 58]. Specifically, we inspire LLMs for divergent thinking by conducting a multimodal debate between two LLM debaters, to generate the rationales derived from harmless and harmful perspectives. Based on the generated harmfulness rationales, we fine-tune a small language model as the debate judge for harmfulness prediction, to align the multimodal features between the meme and the harmfulness rationales. In this manner, our model can effectively focus on contrasting and implicit signals that indicate harmfulness in the debated arguments, while avoiding excessive attention to trivial samples that lack inherent controversy. Our contributions are summarized as follows in three folds:

- To our best knowledge, we are the first to study harmful meme detection from a fresh perspective on harmfulness explainability in natural texts, by harnessing advanced LLMs.[3]
- We propose an explainable approach to conduct a multimodal debate between LLMs on memes for explanation generation from harmless and harmful arguments, which facilitates harmfulness inference with multimodal fusion.
- Extensive experiments conducted on three meme datasets confirm that our universal framework could yield superior performance than previous state-of-the-art baselines for harmful meme detection, and provide informative explanations for better dialectical thinking on meme harmfulness.

---

[3]We will release all the source codes once published.

## 2 RELATED WORK

### 2.1 Harmful Meme Detection

Harmful meme detection is a rapidly growing area in the research community, driven by the recent availability of large meme benchmarks [20, 40, 49]. The Hateful Memes Challenge organized by Facebook [22] further encouraged researchers to develop solutions for detecting harmful memes in hate speech [9]. More recently, Pramanick et al. [40] formally defined the harmful meme concept and demonstrated its dependence on contextual factors. The complex nature of memes, which often rely on multiple modalities, makes them challenging to yield good performance only using unimodal detection methods like BERT [10] or Faster R-CNN [14, 45]. Therefore, recent studies attempted to apply multimodal approaches on the harmful meme detection task.

Previous studies have employed classical two-stream models that integrate text and vision features, which are learned from text and image encoders, typically using attention-based mechanisms and multimodal fusion techniques for classifying harmful memes [20, 22, 49]. Another branch was to fine-tune pre-trained multimodal models specifically for the task [16, 30, 36, 55]. Recent efforts have also sought to explore the use of data augmentation techniques [63, 65], ensemble methods [46, 55, 66] and harmful target disentanglement [26]. Lately, Pramanick et al. [41] proposed a multimodal framework by using global and local perspectives to detect harmful memes, which achieves state-of-the-art performances. The follow-up prompt-based approaches [5, 18] attempted to concatenate the meme text and extracted image captions and fine-tune masked language models [33] for harmful meme detection. A more recent study [4] further improved the image captions with pre-trained vision-language models. However, existing solutions only focused on performing harmful meme classification (*i.e.*, predicting if a meme is harmful) with limited explanations for its prediction [15]. In this paper, we delve into the explainability of harmful meme detection, aiming to convey the accuracy of predictive models using natural language and assisting users in gaining a better understanding.

### 2.2 Large Language Models

LLMs have demonstrated remarkable capabilities in complex reasoning [3, 6, 43, 51], such as generating intermediate inference procedures with CoT prompting before the final output [24, 37, 58, 60]. Advanced sampling strategies have been explored to improve CoT by generating diverse reasoning paths, *e.g.*, Self-Consistency [57], Auto-CoT [60], Complexity-based Consistency [13], Multi-Chain Reasoning [59], and Progressive-Hint Prompting [62]. More recently, some vision LLMs [8, 31, 64] have emerged, showing excellent generalization performance in multimodal tasks. Specifically, LLaVA [31] projects the output of a visual encoder as input to LLaMA [52] and trains both the alignment network and the LLM on synthetic data. Unfortunately, the large size of LLMs restricts their deployment on detecting harmful memes with different modalities, regardless of how they are enhanced with strategic text prompting [61]. In this work, we conduct a multimodal debate between LLMs by using the potential labels as prompting arguments, which further advocates an explainable paradigm to fine-tune smaller language models (LMs) for boosting harmful meme detection.

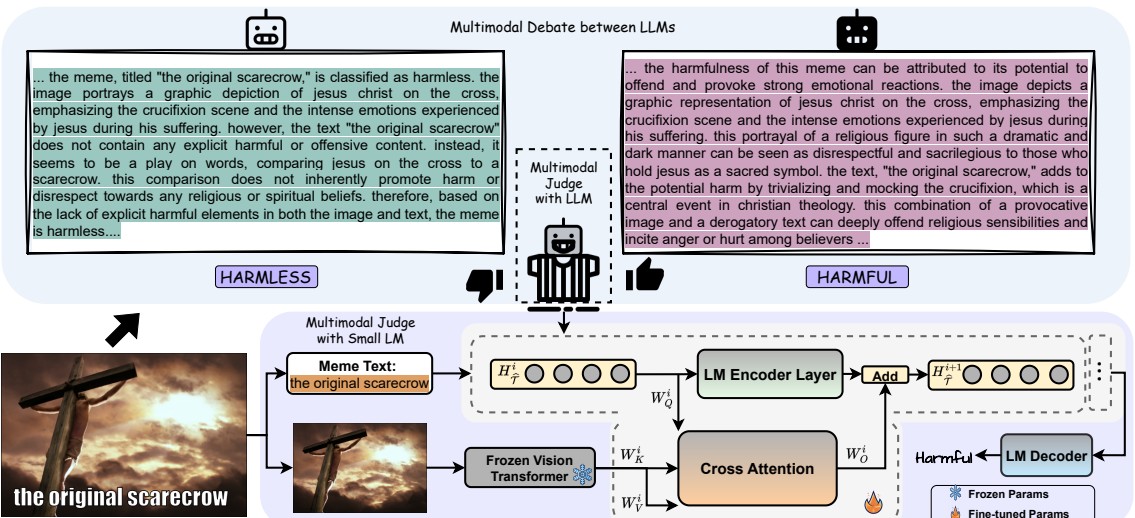

**Figure 2: The overall pipeline of our method. We first conduct the multimodal debate between LLMs, to generate the conflicting rationales from the harmless (green) and harmful (lilac) positions. Then the generated rationales are used to train a small task-specific LM judge with multimodal inputs of memes.**

## 3 OUR APPROACH

### 3.1 Problem Statement

We define a harmful meme detection dataset as a set of memes, where each meme $M = \{\mathcal{I}, \mathcal{T}\}$ is a tuple representing an image $\mathcal{I}$ that is associated with a text sequence $\mathcal{T}$. Following previous work [3, 5, 32], to better transfer and utilize the knowledge learned in pre-trained LMs, this task is formulated as a *natural language generation* problem, where our model takes the meme text $\mathcal{T}$ and the meme image $\mathcal{I}$ as input and generates a textual output of the label $y \in \{\text{harmful}, \text{harmless}\}$ to clearly express whether the meme is harmful or not.

Our core idea is to facilitate reasoning in the model for harmful meme detection with conflicting rationales and dialectical thinking [2], which involves arguments holding different points of view about a subject and strives to arrive at a higher level of resolution, by leveraging LLMs to elicit textual rationales from harmless and harmful perspectives as additional knowledge of the model. The LLM rationale corresponding to the harmfulness prediction of our model could be naturally output as the explanatory basis of the model's decision.

The overview of our framework is shown in Figure 2. It consists of the Multimodal Debate module between LLMs (§3.2), the Multimodal Judge module with LLM (§3.3), and the Multimodal Judge module with Small LM (§3.4).

### 3.2 Multimodal Debate between LLMs

With the aid of LLMs, it becomes plausible to generate natural language rationales that delve into the implicit meaning of memes, facilitating the determination of whether they have harmful implications and serving as a basis to evaluate their harmfulness. However, it is very likely that the rationales generated directly from LLMs can be biased by preconceptions, potentially leading to wrong labels and hampering detection performance [19]. Taking the harmful meme

in Figure 2 as an example, directly prompted with "Is this meme harmless or harmful?", the LLM tends to perceive it as harmless with the rationale that it is a playful comparison between the surface image of Jesus and scarecrows while ignoring the depth of its relevant cultural and religious background. In this paper, we resort to a fundamental characteristic of human problem-solving, *i.e.*, debate, to encourage divergent thinking for harmful meme detection. Fostering the model to explore different viewpoints and reasoning pathways, we design a method to inspire a multimodal debate about the memes between LLMs, in which two agents express their own arguments in the state of "tit for tat" from harmless and harmful perspectives. And then, based on such a bipartisan type of thought chains, a judge agent will be able to infer meme harmfulness by indicating which rationale is more reasonable. In this section, we focus on the prompting method for debate generation.

Given a meme sample $M = \{\mathcal{I}, \mathcal{T}\}$, we curate a prompt template $p^*$ that consists of the meme text $\mathcal{T}$ and the potential harmfulness label $* \in \{hl, hf\}$[4] that denotes "harmless" or "harmful" as observed attributes to prompt the vision LLMs, *i.e.*, LLaVA [31]. Each debater will generate a rationale $r^*$, which elicits the reasoning knowledge about how to infer a given harmfulness label $*$ based on the interplay of the meme text $\mathcal{T}$ and the meme image $\mathcal{I}$.

Specifically, we design the prompt template $p^*$ as follows:

"*Given the meme, with the Text: [$\mathcal{T}$] embedded in the Image, please provide a streamlined explanation associated with the text and the image by using the contextual background commonsense knowledge, to explicitly explain how the harmfulness of the meme is reasoned as [$*$].*".

Based on the template, each LLaVA debater employs a variant of texts, *i.e.*, either $p^{hl}$ or $p^{hf}$, as the prompt to generate a corresponding rationale $r^{hl}$ or $r^{hf}$, derived from the harmless and harmful positions, respectively. As we provide the potential harmfulness

---

[4]Here the potential harmfulness labels are just used to formalize two opposite standpoints to argue regardless of what the ground-truth label is.

label as part of the attributes in each specific prompt, the rich contextual background knowledge could be activated to generate a rationale for supporting the argument that intends to promote the potential harmfulness label separately in each debate. In this way, the contextual nuances of memes that contribute to the respective candidate harmfulness categories could be better presented and contrasted, so that the true harmfulness will be revealed and reasoned by the rest of the model from dialectical views.

## 3.3 Multimodal Judge with LLM

Inspired by the CoT prompting [58], we exploit the competing arguments from the multimodal debate to provide dialectical reasoning chains for an LLM judge with emergent abilities, enabling it to decide whether memes are harmful or not. Thus, we feed the conflicting rationales as reasoning steps into the LLM judge for inferring the predicted harmfulness label.

While the judge can be implemented using any LLM with comparable or even stronger capabilities as the debaters, we opt to employ the same LLM for the judge to ensure that no additional variates, such as different kinds of inductive biases, are introduced. Specifically, we cast the rationales $r^{hl}$ and $r^{hf}$ into a conflicting thought chain using the CoT prompting method [58, 60], acting as the textual prompt of the LLM judge:

"*Given the meme, with the Text: [$\mathcal{T}$] embedded in the Image, and the following two meme rationales: (1) Harmless: [$r^{hl}$]; (2) Harmful: [$r^{hf}$], is this meme harmless or harmful?*"

Following the input prompt, the LLM judge infers the harmfulness label, which actually provides its *preference* over the two labels indicating which corresponding explanation from the debate is more reasonable than the other one. This could be either taken as the final output alone, or used as an extra reference for a more accurate prediction model described in the subsequent section.

## 3.4 Multimodal Judge with Small LM

Although the LLM judge is enhanced by the multifaceted information provided from the multimodal debate, its inference still could be unreliable due to the inherent limitations of LLMs [1, 17]. On the other hand, it is impractical to fine-tune the LLM judge for this task due to the huge amount of model parameters. For a more reliable judgment, we propose to fine-tune a smaller LM judge that classifies memes as harmful or harmless, by leveraging the rationales derived from the contradictory harmfulness arguments as prior knowledge. This design strives to facilitate multimodal interactions, allowing the rationales from the LLM debaters to effectively synergize with the intrinsic multimodal information present in memes.

For a meme sample $M = \{\mathcal{I}, \mathcal{T}\}$, we first concatenate the meme text $\mathcal{T}$ and the harmfulness rationales as the input text of our Small LM judge. Similar to the fixed input order of rationales in the LLM judge, we initialize the input text $\widehat{\mathcal{T}}$ of the Small LM judge as:

$$\widehat{\mathcal{T}} = [\mathcal{T}, r^{hl}, r^{hf}], \tag{1}$$

where $[\cdot, \cdot, \cdot]$ denotes the concatenation operation.

Alternatively, we can refer to the harmfulness inference result given by the LLM judge when fine-tuning the Small LM judge. In this setting, we place the one rationale that the LLM judge deems more reasonable in front of the other one. Specifically, we prepare

the input text $\widehat{\mathcal{T}}$ as:

$$\widehat{\mathcal{T}} = [\mathcal{T}, r^{(1)}, r^{(2)}], \tag{2}$$

where $(1) \succ (2)$, denoting that the LLM judge prefers $r^{(1)}$ to $r^{(2)}$, and $(1), (2) \in \{hl, hf\}$. Compared to a fixed sequence of rationales in Equation 1, adjusting the rationale order based on the prior from the LLM judge aims to implicitly leverage LLM knowledge and insights. This adjustment helps the Small LM judge prioritize challenging training examples that were misjudged by the LLM judge, while avoiding excessive attention to trivial examples that have already been correctly detected by the LLM judge. By standing upon the shoulders of giants, we hypothesize that the model can better refine its understanding of the memes while learning to rectify the misperception of the LLM judge with the training data.

Then we encode the input text $\widehat{\mathcal{T}}$ and the meme image $\mathcal{I}$ to obtain their embedding vectors as follows:

$$H^0_{\widehat{\mathcal{T}}} = \text{TE}(\widehat{\mathcal{T}}), \ H_{\mathcal{I}} = \text{VE}(\mathcal{I}), \tag{3}$$

where $\text{TE}(\cdot)$ denotes the text embedding layer of the LM Encoder. And $H^0_{\widehat{\mathcal{T}}} \in \mathbb{R}^{m \times d}$ is the token embeddings output by the embedding layer of Transformer encoder [54], where $m$ is the text length of $\widehat{\mathcal{T}}$ and $d$ is the size of the hidden states. Benefiting from the Relative Position Encoding of LMs [44], the judgment by LLM could be injected into the Small LM judge based on the relative position information of the input sequence. $\text{VE}(\cdot)$ is the Vision Extractor based on a pre-trained vision Transformer [42] with frozen parameters. It is used to fetch the patch-level features of the image with $n$ patches, which is projected into the visual representations $H_{\mathcal{I}} \in \mathbb{R}^{n \times d}$.

To support semantic alignment between the meme sample and the harmfulness rationales for better cross-modal context understanding, we exploit a cross-attention mechanism to attend the visual representations to the textual ones, for Multimodal Fusion of the textual and visual information in our Small LM judge:

$$H^i_{\mathcal{I}} = \text{softmax}\left(\frac{Q_{\widehat{\mathcal{T}}} K^\top_{\mathcal{I}}}{\sqrt{d_k}}\right) V_{\mathcal{I}}, \tag{4}$$

where the query, key and value are defined as $\{Q_{\widehat{\mathcal{T}}}, K_{\mathcal{I}}, V_{\mathcal{I}}\} = \{H^i_{\widehat{\mathcal{T}}} W^i_Q, H_{\mathcal{I}} W^i_K, H_{\mathcal{I}} W^i_V\}$, $\{W^i_Q, W^i_K, W^i_V\} \in \mathbb{R}^{d \times d_k}$ are trainable weights, $H^i_{\widehat{\mathcal{T}}}$ is the input hidden states of the $i$-th LM Encoder layer and $H^i_{\mathcal{I}}$ is the attended visual features. Then we can fuse $H^i_{\mathcal{I}}$ with $H^i_{\widehat{\mathcal{T}}}$ to attain the interplay representations for a meme:

$$H^{i+1}_{\widehat{\mathcal{T}}} = \text{LME}^i\left(H^i_{\widehat{\mathcal{T}}}\right) + H^i_{\mathcal{I}} W^i_O, \tag{5}$$

where $\text{LME}^i(\cdot)$ is the $i$-th layer of the LM Encoder, $W^i_O$ denotes the linear projection, and $0 \le i \le L-1$ given the total $L$ layers in the LM Encoder. We denote $\widehat{H} = H^L_{\widehat{\mathcal{T}}}$ as the final interplay representations.

*Model Training.* We feed the interplay representations $\widehat{H} \in \mathbb{R}^{m \times d}$ into the LM Decoder, implemented as a Transformer-based decoder, to generate the predicted label. With the generative objective [44] adapted to pre-trained LMs, the Small LM judge could leverage prior reasoning knowledge absorbed in the pre-training stage to better deduce harmfulness prediction. Specifically, our Small LM

**Table 1: Harmful meme detection results on three datasets. The accuracy and macro-averaged F1 score (%) are reported as the metrics. The best and second test results are in bold and underlined, respectively. ($p < 0.05$ under t-test)**

| Dataset | Harm-C | | Harm-P | | FHM | |
|---|---|---|---|---|---|---|
| Model | Accuracy | Macro-$F_1$ | Accuracy | Macro-$F_1$ | Accuracy | Macro-$F_1$ |
| Text BERT [10] | 70.17 | 66.25 | 80.12 | 78.35 | 57.12 | 41.52 |
| Image-Region [14] | 68.74 | 62.97 | 73.14 | 72.77 | 52.34 | 34.19 |
| Late Fusion [40] | 73.24 | 70.25 | 78.26 | 78.50 | 59.14 | 44.81 |
| MMBT [20] | 73.48 | 67.12 | 82.54 | 80.23 | 65.06 | 61.93 |
| VisualBERT COCO [28] | 81.36 | 80.13 | 86.80 | 86.07 | 61.48 | 47.26 |
| ViLBERT CC [34] | 78.70 | 78.09 | 87.25 | 86.03 | 64.70 | 55.78 |
| MOMENTA [41] | 83.82 | 82.80 | 89.84 | 88.26 | 61.34 | 57.45 |
| MaskPrompt [5] | 84.47 | 81.51 | 88.17 | 87.09 | 72.98 | 65.24 |
| Pro-Cap [4] | 85.01 | 83.17 | 89.32 | 87.91 | 74.95 | 71.68 |
| ExplainHM | **87.00** | **86.41** | **90.73** | **90.72** | **75.60** | **75.39** |

judge denoted as $f(\mathcal{I}, \widehat{\mathcal{T}})$ is trained by minimizing the loss:

$$\mathcal{L} = \text{CE}\left(f(\mathcal{I}, \widehat{\mathcal{T}}), y\right), \tag{6}$$

where $\text{CE}(\cdot)$ denotes the cross-entropy loss [50] between the generated label token and the ground-truth harmfulness label $y$.

When the LLM judge is not considered, the relative positions between the harmfulness rationales are invariant. Thus, the order information might not affect the model's learning much. When integrated with the LLM judge, our model can be aware of the variation of relative positions between the harmfulness rationales given the prior preference of the LLM judge, which would encourage the model to learn by contrasting with the ground-truth labels. To this end, we utilize the T5 encoder-decoder architecture [7, 44] with Relative Position Encoding to initialize our model. In this manner, during the task-specific fine-tuning process, our Small LM judge is able to attend over the implicit harm-indicative patterns in the rationales that were incorrectly inferred by the LLM judge, thus improving the overall detection performance. Meanwhile, the rationale indicated by the final prediction could serve as a supportive basis to explain the decision in natural language.

## 4 EXPERIMENTS

### 4.1 Experimental Setup

**Datasets.** We use three publicly available meme datasets for evaluation: (1) Harm-C [40], (2) Harm-P [41], and (3) FHM [22]. Harm-C and Harm-P consist of memes related to COVID-19 and US politics, respectively. FHM was released by Facebook as part of a challenge to crowd-source multimodal harmful meme detection in hate speech solutions. Different from FHM that each meme was labeled as *harmful* or *harmless*, Harm-C and Harm-P were originally labeled with three classes: *very harmful*, *partially harmful*, and *harmless*. For a fair comparison, we merge the *very harmful* and *partially harmful* memes into the *harmful* class, following the setting of recent work [4, 5, 41].

**Baselines.** We compare our model with several state-of-the-art (SoTA) harmful meme detection systems: 1) **Text BERT** [10]; 2) **Image-Region**[14, 45]; 3) **Late Fusion** [40]; 4) **MMBT** [20]; 5) **VisualBERT COCO** [28, 29]; 6) **ViLBERT CC** [34]; 7) **MOMENTA** [41]; 8) **MaskPrompt** [5]; 9) **Pro-Cap** [4]. We use accuracy and macro-averaged F1 score as the evaluation metrics.

The data statistics, baseline descriptions and model implementation are detailed in Appendix §A, §B and §C, respectively.

### 4.2 Harmful Meme Detection Performance

Table 1 demonstrates the performance of our proposed method ExplainHM versus all the compared harmful meme detection methods on the Harm-C, Harm-P and FHM datasets. It is observed that 1) The performance of the baselines in the first group is significantly lower, primarily because they only utilize unimodal features such as either text or image. On the other hand, the remaining baselines effectively leverage the multimodal features extracted from both text and image parts of memes. 2) The multimodal models in the second group outperform the unimodal ones. The early-fusion models with multimodal pre-training (*i.e.*, VisualBERT COCO and ViLBERT CC) outperform the simple fusion with unimodal pre-training (*i.e.*, Late Fusion and MMBT) on Harm-C and Harm-P datasets, while MOMENTA performs relatively better in the second group by considering global and local information of memes, especially on the Harm-P dataset. 3) However, as the images in FHM datasets are more informative and high-quality, MaskPrompt outperforms MOMENTA by incorporating additional extracted entities and demographic information of the image into the masked language models, besides just captioning the image into the prompt. Based on MaskPrompt, Pro-Cap further improves image captioning with pre-trained vision-language models [27], which leads to the best performance among all the baselines.

Under the full setting (*i.e.*, with the integration of the LLM judge and Small LM judge), our ExplainHM improves over the best baselines by 3.24%, 2.46%, and 3.71% in terms of Macro-F1 score on Harm-C, Harm-P, and FHM datasets, respectively. We observe that 1) The improvements observed on the Harm-P dataset are relatively subdued compared to the advancements made on the other two datasets. Moreover, there are minimal discrepancies in the performance of all the baselines on the Harm-P dataset. This can be attributed to the scale of the Harm-P dataset, which not only has the smallest volume of data but also exclusively comprises politics-related harmful memes. 2) A similar phenomenon is evident in the Harm-C and FHM datasets, where ExplainHM demonstrates greater performance improvements as the scale and the difficulty of the dataset increase. ExplainHM showcases consistent and adaptable performance across all benchmark datasets for harmful meme

**Table 2: Ablation studies by removing components from our proposed framework.**

| Dataset | Harm-C | | Harm-P | | FHM | |
|---------|--------|--------|--------|--------|-------|--------|
| Model | Acc. | Mac-$F_1$ | Acc. | Mac-$F_1$ | Acc. | Mac-$F_1$ |
| ExplainHM | 87.00 | 86.41 | 90.73 | 90.72 | 75.60 | 75.39 |
| w/o MD | 83.33 | 81.44 | 88.17 | 88.17 | 73.60 | 73.41 |
| w/o LLMJ | 85.59 | 84.80 | 88.44 | 88.43 | 71.40 | 70.94 |
| w/o SLMJ | 58.19 | 56.17 | 56.11 | 52.59 | 57.00 | 56.61 |
| w/o HlD | 82.20 | 81.76 | 89.06 | 89.05 | 70.60 | 69.55 |
| w/o HfD | 85.31 | 84.78 | 88.75 | 88.74 | 72.60 | 72.26 |
| w/o MF | 83.90 | 83.30 | 88.44 | 88.43 | 72.80 | 72.20 |
| w/o UR | 85.59 | 84.80 | 89.38 | 89.37 | 73.40 | 73.24 |

**Table 3: Ablation studies by adding paradigms on LLMs.**

| Dataset | Harm-C | | Harm-P | | FHM | |
|---------|--------|--------|--------|--------|-------|--------|
| Model | Acc. | Mac-$F_1$ | Acc. | Mac-$F_1$ | Acc. | Mac-$F_1$ |
| LLaVA | 50.28 | 49.70 | 49.84 | 34.86 | 51.20 | 46.51 |
| w/ MD_CoT | 58.19 | 56.17 | 56.11 | 52.59 | 57.00 | 56.61 |
| w/ ExplainHM | 87.00 | 86.41 | 90.73 | 90.72 | 75.60 | 75.39 |
| ChatGPT | 70.06 | 64.05 | 59.87 | 58.02 | 56.20 | 55.50 |
| w/ MD_CoT | 68.08 | 65.82 | 62.07 | 61.69 | 62.00 | 61.62 |
| w/ ExplainHM | 86.16 | 85.22 | 90.00 | 89.98 | 76.60 | 76.39 |

detection, thanks to its astute discernment of harmful memes. The key differentiator lies in the fact that while all the baselines solely focus on recognition, our model is equipped with rationales from multimodal debate, which empowers our model to unveil harmful content by leveraging seemingly unrelated textual and visual elements within memes.

## 4.3 Ablative Studies

We perform ablative studies on several variants of ExplainHM: 1) *w/o Multimodal Debate (MD)*: Simply fine-tune the Smaller LM judge with the multimodal fusion of the meme text and the meme image without the stage of multimodal debate between LLMs; 2) *w/o LLM Judge (LLMJ)*: Simply concatenate the harmfulness rationales into the input text in a fixed order as Equation 1 without pre-ranked by the LLM judge; 3) *w/o Small LM Judge (SLMJ)*: Simply use the output of the LLM judge as the final prediction, as depicted in §3.3; 4) *w/o Harmless Debater (HlD)*: Only concatenate the rationale from the harmful argument together with the meme text as the input text of the Small LM judge; 5) *w/o Harmful Debater (HfD)*: Only concatenate the rationale from the harmless argument together with the meme text as the input text of the Small LM judge; 6) *w/o Multimodal Fusion (MF)*: Instead of the fusion mechanism on the multimodal features in our Small LM judge, we only append the linguistic features from image captioning together with the input text during encoding; 7) *w/o Unpreferred Rationale (UR)*: Only concatenate the rationale preferred by the LLM judge and the meme text as the input text of the Small LM judge.

As demonstrated in Table 2, the ablative models suffer different degrees of performance degradation, indicating the effectiveness of our proposed components for harmful meme detection by multimodal debate between LLMs and multimodal fusion with small LM. Specifically, the performance of ExplainHM largely decreases

in the 'w/o MD' setting due to the lack of multimodal rationales generated from LLMs about the seemingly uncorrelated modalities in memes. The 'w/o LLMJ' setting also achieves worse performance than ExplainHM, suggesting that the prior preference of the LLM judge on the rationales from different positions plays an important role and provides positive guidance in identifying the harm-indicative elements in memes. For 'w/o SLMJ', the decrease is significant, underscoring the importance of the Small LM judge fine-tuned specifically for this task. ExplainHM makes improvements over 'w/o HlD' and 'w/o HfD', which implies the promoting role of our multimodal debate mechanism that incorporates rationales from the harmless and harmful arguments into the language model. Moreover, the 'w/o HlD' setting leads to a larger performance drop than 'w/o HfD', because the amount of the harmless meme samples in the training data is more than that of the harmful ones. Compared with ExplainHM, the performance of 'w/o MF' also significantly decreases, highlighting the importance of the cross-attention fusion mechanism to mitigate the possible misalignments, like the information loss about the meme images in the rationales. In the 'w/o UR' setting, we further remove the rationale not preferred by the LLM judge in the input text of the Small LM judge, which also results in performance degradation. This reaffirms the usefulness of the conflicting rationales appended in the input text that make our model hardly compromised when there could be discrepancies between the LLM judge and the ground truth.

To enhance the robustness of the detection performance evaluation, we further conduct the ablative studies by adding the paradigms on LLMs to draw more insightful comparisons among variants of LLMs, as shown in Table 3. LLaVA and ChatGPT are selected as the representative LLMs from the vision and language perspectives. We devise three variants of paradigms based on LLMs for the harmful meme detection task: 1) *LLaVa/ChatGPT*: Directly prompt a representative LLM, to infer harmfulness for harmful meme detection; 2) *w/ MD_CoT*: The LLM judge with Multimodel Debate CoT reasoning but without the presence of Small LM judge, the similar setting to 'w/o SLMJ' in Table 2; 3) *w/ ExplainHM*: Our proposed paradigm ExplainHM under full setting based on the integration of the LLM judge and Small LM judge, where LLMs are LLaVA or ChatGPT.

We have the following observations: 1) The direct deployment of both LLaVA and ChatGPT struggles since the models are not specifically designed for this task, highlighting the necessity of our Multimodal Debate mechanism to alleviate the issues of directly promoting LLMs for harmfulness prediction. 2) The 'w/ MD_CoT' prompting strategy could effectively enhance the detection performance of LLMs, especially LLaVA, which suggests that the conflicting rationale generation from the Multimodal Debate stage is a reasonable way to optimize the reasoning chains for LLMs applied to the harmful meme detection task. 3) Besides using the 'w/ MD_CoT' prompting strategy in the LLM judge, our proposed paradigm 'w/ ExplainHM' further improves the model's performance by focusing on the fine-tuning of the Small LM judge to avoid impractical fine-tuning of the LLM judge while considering the prior preference given by the LLM judge. Furthermore, the 'w/ ExplainHM' setting achieves excellent performance based on both LLaVA and ChatGPT, which demonstrates that the choice of LLMs is orthogonal to our proposed paradigm that can be easily augmented with existing LLMs without any other change.

**Table 4: Automatic GPT-4 evaluation of the explanation quality on harmful memes in FHM dataset.**

| Explanations | LLaVA | ChatGPT | Human |
|---|---|---|---|
| Informativeness | 4.07 | 4.94 | 2.27 |
| Readability | 4.71 | 4.98 | 2.36 |
| Soundness | 4.25 | 4.87 | 2.99 |
| Conciseness | 3.93 | 3.19 | 4.07 |
| Persuasiveness | 4.03 | 4.82 | 2.64 |

**Table 5: Human evaluation of the explanation quality on harmful memes in FHM dataset.**

| Explanations | LLaVA | ChatGPT | Human |
|---|---|---|---|
| Informativeness | 4.05 | 4.01 | 2.64 |
| Readability | 3.99 | 3.95 | 3.96 |
| Soundness | 3.81 | 3.97 | 2.94 |
| Conciseness | 3.25 | 3.12 | 4.30 |
| Persuasiveness | 3.75 | 3.76 | 2.78 |

## 4.4 Evaluation of Explainability

**Automatic Evaluation.** Generally, there is no gold explanation about memes for the harmful meme detection task due to the diverse forms of textual expression. Devising reliable metrics without reference is not a straightforward task and can also be problematic. Furthermore, different types of text necessitate the evaluation of distinct aspects, such as informativeness, fluency, soundness, etc. [11, 35], which makes it hard to design metrics for each type of text and dimension separately. Nowadays, GPT-4 [38] has revolutionized the field of LLMs with a more powerful expressive capacity. In this subsection, we present a new automatic evaluation using GPT-4 in a reference-free mode, to evaluate the text quality of the explanations generated by our approach from LLaVA and ChatGPT.

We randomly selected 3,000 harmful samples from the FHM dataset. For a more comprehensive comparison, we further provide GPT-4 with human-written explanations in hate speech by drawing the practice of previous literature [15] for the sampled memes. Specifically, GPT-4 is prompted to score the explanations w.r.t. each meme according to the following criteria: 1) *Informativeness*: the explanation provides new information, such as explaining the background and additional context; 2) *Readability*: the explanation follows proper grammar and structural rules; 3) *Soundness*: the explanation seems valid and logical; 4) *Conciseness*: the explanation contains less redundant information; 5) *Persuasiveness*: the explanation seems convincing. For each criterion, a 5-point Likert scale was employed, where 1 meant the poorest quality and 5 the best.

Table 4 demonstrates the averaged scores of the explanation evaluation by GPT-4 on the three sources (*i.e.*, LLaVA, ChatGPT, and Human) regarding the five criteria. We could observe that: 1) Compared with LLaVA and ChatGPT, the explanations written by human beings [15] are generally scored the highest in Conciseness but the lowest in the other aspects, because the mean explanation length is 13.62 which is shorter than that of LLaVA (125.37) and ChatGPT (180.82). 2) Interestingly, although GPT-4 is more powerful than LLaVA and ChatGPT, it tends to give higher scores to ChatGPT overall than LLaVA. We speculate the reason for such a bias is that both GPT-4 and ChatGPT are developed as successors of the LLM

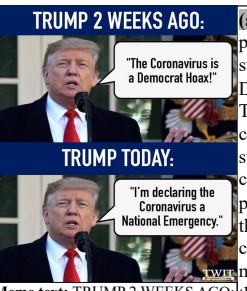

**(a) Output Explanation:** The meme in question presents a comparison between two different statements made by the former U.S. President, Donald Trump. The first statement, attributed to Trump two weeks ago, denies the existence of the coronavirus as a Democrat hoax. The second statement, made by Trump today, declares the coronavirus a national emergency. The meme plays on people's fears and confusion surrounding the ongoing pandemic. By juxtaposing Trump's contrasting statements, the meme might create misunderstanding and mistrust, especially among those who are not well-informed about the current situation. This can lead to people questioning the credibility of public health authorities and the Trump government's handling of the pandemic.

**Meme text:** TRUMP 2 WEEKS AGO: "The Coronavirus is a Democrat Hoax!" TRUMP TODAY: "I'm declaring the Coronavirus a National Emergency."

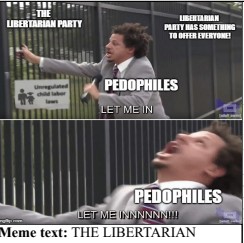

**(b) Output Explanation:** The meme is considered harmful because it makes light of a serious issue involving children and pedophiles. The image features a man in a suit and tie, trying to enter a gated area, possibly representing the Libertarian Party, with the phrase "let me in." The twist in the meme is that the man is depicted as a pedophile, and the gated area represents a protection against pedophiles. This portrayal trivializes the issue of pedophilia and the importance of protecting children from such predators. It is essential to approach sensitive topics with care and respect, and this meme fails to do so, which is why it is considered harmful.

**Meme text:** THE LIBERTARIAN PARTY. LIBERTARIAN PARTY HAS SOMETHING TO OFFER EVERYONE! Unregulated child labor laws. PEDOPHILES. LET ME IN PEDOPHILES. LET ME INNNNNN!!!

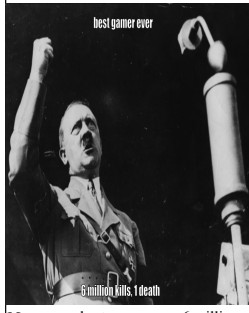

**(c) Output Explanation:** The meme featuring the image of a man in a military uniform and the text 'best gamer ever 6 million kills, 1 death' is considered harmful because it trivializes and makes light of a serious and tragic historical event - the Holocaust. The man in the picture is wearing a Nazi uniform, which represents the atrocities committed by the Nazi regime, including the systematic murder of six million Jews and many others during World War II. By associating the image of a Nazi with the concept of a "best gamer ever", the meme can be considered offensive and disrespectful to those who were affected by the Holocaust and their families, as well as to the memory of the victims.

**Meme text:** best gamer ever 6 million kills, 1 death

**Figure 3: Examples of correctly predicted harmful memes in (a) Harm-C, (b) Harm-P, and (c) FHM datasets.**

InstructGPT [39], so that the generated explanations by ChatGPT is more to the taste of GPT-4. 3) The performance of LLaVA evaluated by GPT-4 achieves an excellent balance between Conciseness and Persuasiveness, which implies that the LLaVA-generated explanations could succinctly impress GPT-4. We provide more results on the Harm-C and Harm-P data, and details on the reproducibility of the automatic GPT-4 evaluation in Appendix §G and §C.5.

**Human Evaluation.** Considering that automatic evaluation cannot realistically measure the quality of the chosen explanations generated by the multimodal debate between LLMs, we further conduct the human subjects study to evaluate the overall quality of explainability. 50 harmful samples are randomly selected from the FHM test set and 10 professional linguistic annotators are asked to evaluate the explanations of our model from LLaVA [31] and ChatGPT [39], further with those written by Human [15]. The metrics of human evaluation are the same as the automatic evaluation.

The scores of human evaluation are shown in Table 5. Note that the intra-class agreement score is 0.625. The average Spearman's

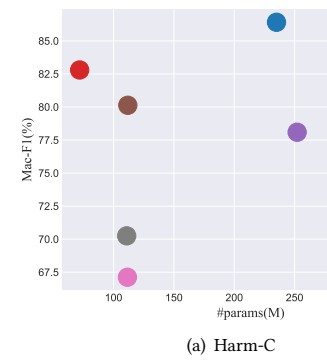
(a) Harm-C

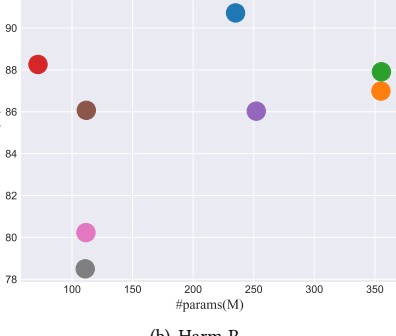
(b) Harm-P

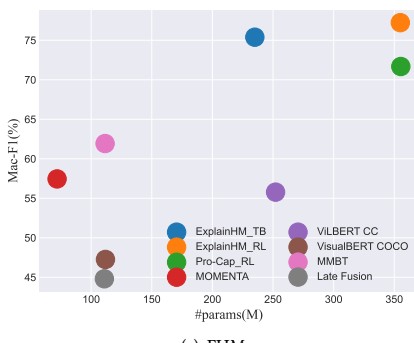
(c) FHM

Figure 4: The performance of our EXPLAINHM and other multimodal baselines with respect to the parameter size.

correlation coefficient between any two annotators is 0.682. We can observe that: 1) The readability scores of all sources are high, which is because LLMs can generate fluent sentences as human beings towards the harmful meme explanation with commonsense knowledge. Compared with the automatic GPT-4 evaluation, the readability score of human-written explanations largely improved in human evaluation. 2) Except for the readability scores, the scores of human-written explanations in the other four metrics are similar in both human evaluation and automatic GPT-4 evaluation. 3) It is worth noting that although the explanations generated by our framework from the multimodal debate on ChatGPT received relatively lower scores in the human evaluation compared to those under automatic GPT-4 evaluation, it still, along with LLaVA, both achieved an overall superior performance to the human-written explanations from previous work [15], which do not explicitly explain the complex and integrated semantic information present in the memes. 4) According to the feedback of the evaluators, LLaVA can efficiently generate more concise explanations with correct key contents than ChatGPT which tends to generate lengthy and inclusive sentences. Overall, although there is still room for developing a more comprehensive metric for evaluating harmfulness explanations, the results reveal that it is feasible and reasonable for us to devise such a universal framework for harmful meme detection and explanation, by leveraging the impressive abilities of text generation in LLMs. More human subject studies and broader ethical statements are provided in Appendix §H and §L, respectively.

## 4.5 Case Study

One key advantage of our model is that the rationales generated in the multimodal debate between LLMs could serve as the output explanations for predicted results. For the correctly predicted harmful meme test samples, the output explanation refers to the rationale from the harmful argument, to understand the model predictions more transparently and intuitively, as exemplified in Figure 3.

From the explanations in natural text, we observe that 1) the multimodal information related to the meme text and image could be well understood with commonsense knowledge. For example, in Figure 3(a), the character in the image is recognized as "the former U.S. President", which could be linked to the "TRUMP" in the text; in Figure 3(b), the recognized "gated area" in the image could be recognized as protection against "PEDOPHILES" to satire "THE LIBERTARIAN PARTY" in the text; and in terms of Figure 3(c), the

man in the image could be associated with "the Nazi regime" related to "gamer" in the text. 2) Furthermore, the interplay of multimodal information could be cognized with advanced reasoning. Benefitting from the rich multimodal understanding of the memes, the "comparison between two different statements" in Figure 3(a) can be reasoned to cause harmful consequences like "misunderstanding and mistrust"; the juxtaposition of a political party with the harmful topics "involving children and pedophiles" could be reasoned as trivializing such a serious issue in Figure 3(b); and the meme in Figure 3(c) shows disrespects to "those who were affected by the Holocaust and their families". In this way, the rich but implicit correlations between the meme text and image could be explained in readable snippets, which are also potentially valuable for aiding human checkers to verify the model predictions. We further provide more case studies and error analysis in Appendix §J and §I.

## 4.6 Impact of Model Size and Backbones

We provide a comparison of performance on the three meme datasets with regard to the number of trainable parameters for EXPLAINHM and the other multimodal baselines in Figure 4. We can observe that our model (EXPLAINHM_TB) has already achieved outstanding performance on the three benchmarks with T5$_{Base}$ as the Small LM judge, which has a smaller size than the SoTA baseline Pro-Cap_RL based on RoBERTa$_{Large}$ (over 300M parameters). For a more comprehensive comparison, we revise our Small LM judge with the backbone of Pro-Cap_RL to build the EXPLAINHM_RL model. We observed that EXPLAINHM_RL still outperforms all the baselines on Harm-C and FHM datasets by a large margin, yet is competitive on Harm-P due to the smaller data scale. In general, the results demonstrate that EXPLAINHM is effective and not overly dependent on the parameter size of the Small LM judge to enhance performance.

## 5 CONCLUSION AND FUTURE WORK

We proposed an explainable approach for harmful meme detection. We first conducted a multimodal debate between LLMs about the meme to generate contradictory rationales from harmless and harmful arguments. Then utilizing these rationales, we designed a tunable language model as the judge to infer meme harmfulness. Our proposed framework is evaluated on three meme benchmarks, demonstrating its effectiveness in both detection and explainability. Moving forward, we plan to further enhance the automatic evaluation of the explanation quality as part of our future work.

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

**Table 6: Statistics of Datasets.**

| Datasets | Train | | Test | |
|---|---|---|---|---|
| | #harmful | #harmless | #harmful | #harmless |
| Harm-C | 1064 | 1949 | 124 | 230 |
| Harm-P | 1486 | 1534 | 173 | 182 |
| FHM | 3050 | 5450 | 250 | 250 |

## A DATASETS AND METRICS

The statistics of the three datasets are shown in Table 6. We use the accuracy and macro-averaged F1 score as the evaluation metrics of harmful meme detection, where the macro-averaged F1 is the more important metric owing to the imbalanced class prevalence (see Table 6), to capture competitive performance beyond the majority class.

## B BASELINES

We compare our model with several state-of-the-art harmful meme detection systems: 1) **Text BERT**: BERT [10] is utilized as the unomodal text-only model; 2) **Image-Region**: a unimodal visual-only model that processes meme images using Faster R-CNN [45] with ResNet-152 [14] to feed into a classification layer; 3) **Late Fusion**: a multimodal model uses the average prediction scores of BERT and ResNet-152 for harmful meme detection [40]; 4) **MMBT**: a multimodal Bi-Transformer [20] that captures the intra-modal and inter-modal dynamics of the two modalities; 5) **VisualBERT COCO**: Visual BERT [28] pre-trained on the COCO dataset [29]; 6) **ViLBERT CC**: Vision and Language BERT [34] trained on an intermediate multimodal objective [47] for task-agnostic joint representations of image and text; 7) **MOMENTA**: a multimodal harmful meme detection system [41] that takes the global and local information in two modalities of memes into account; 8) **MaskPrompt**: a prompt learning approach [5] that concatenates the meme text and the image caption as the prompt for masked language modeling [33]; 9) **Pro-Cap**: a caption-enhanced version [4] of MaskPrompt, by leveraging pre-trained vision-language models with probing queries, to improve the image caption in the text prompt.

## C IMPLEMENTATION DETAILS

### C.1 Prompting LLaVA for Multimodal Debate and Judge

As depicted in §3.2 and §3.3, we utilize the vision LLM, *i.e.*, LLaVA [31], specifically the "llava-13b-v1-1" version[5] as the implementation of the debaters and the judge. The detailed prompting design is exemplified in §D. To ensure our results are reproducible, we set the temperature as 0 and the maximum length as 256 without any sampling mechanism.

### C.2 Prompting ChatGPT Debaters for Multimodal Debate

We have introduced the prompting design for LLaVA to conduct the multimodal debate. Here we would introduce how to prompt

---

[5]https://huggingface.co/liuhaotian/LLaVA-13b-delta-v1-1

ChatGPT [39], a widely used LLM developed by OpenAI, specifically utilizing the "gpt-3.5-turbo" version, as another variant of our approach in the ablative studies. To prompt ChatGPT for the multimodal debate, we need to convert the meme's image into an acceptable textual input for ChatGPT. We first in-paint the memes by combining MMOCR [25] with SAM [23] to extract the text and pure image in memes. Then for a fair comparison with LLaVA, we apply LLaVA to generate textual descriptions about the dominant objects or events in the memes' image. To make our results reproducible, we set the temperature as 0 and the maximum length as 256 in ChatGPT.

For the system prompt to ChatGPT Debaters, we design the message as follows:

"*You have been specially designed to perform abductive reasoning for the harmful meme detection task. Your primary function is that, according to a harmfulness label about an image with a text embedded, please provide a streamlined rationale, without explicitly indicating the label, for how it is reasoned as the given harmfulness label. The image and the textual content in the meme are often uncorrelated, but its overall semantic information is presented holistically. Thus it is important to note that you are prohibited from relying on your own imagination, as your goal is to provide the most accurate and reliable rationale possible so that people can infer the harmfulness according to your reasoning about the background context and relationship between the given text and image caption.*".

Moreover, to prompt ChatGPT to generate reasonable rationales with the triplet $\{*, \tilde{\mathcal{I}}, \mathcal{T}\}$ as observed attributes, where $\tilde{\mathcal{I}}$ is the image caption, we design the template $p^*$ for the user prompt as:

"*Given a Text: [$\mathcal{T}$], which is embedded in an Image: [$\tilde{\mathcal{I}}$]; and a harmfulness label [*], please give me a streamlined explanation associated with the meme, without explicitly indicating the label, to explain how the harmfulness of the meme is reasoned as [*].*".

## C.3 Prompting ChatGPT to Judge Multimodal Debate

For prompting a ChatGPT Judge, we design the message as follows:

"*Suppose you are a harmful meme detector. There will be two debaters involved in a meme explanation competition for harmful meme detection. As there must be a winner and a loser in a debate, your job is to determine whether a meme is harmful or harmless by evaluating their stated explanations on different positions (harmful or harmless). The image and the text may not be harmful when considered in isolation, but are harmful when taken as a whole and vice versa. Thus it is important to note that you are prohibited from relying on your own imagination and being affected by exaggerated or misinterpreted explanations, as your goal is to provide the most accurate and reliable judgment possible.*".

Moreover, to prompt ChatGPT to make a judgment with the tuple $\{\tilde{\mathcal{I}}, \mathcal{T}\}$ as observed attributes, where $\tilde{\mathcal{I}}$ is the image caption, we design the user prompt as:

"*Given a Text: [$\mathcal{T}$], which is embedded in an Image: [$\tilde{\mathcal{I}}$]; with the following two rationales: (1) Harmless: [$r^{hl}$]; (2) Harmful: [$r^{hf}$], is this meme harmless or harmful?*".

For the input order of the harmless and harmful rationales, we found there is not much difference between the judgment results for

**Table 7: Hyper-parameters.**

| Hyper-Parameter | Harm-C | Harm-P | FHM |
|---|---|---|---|
| epoch | 20 | 20 | 20 |
| batch size | 32 | 32 | 32 |
| Learning Rate | 5e-5 | 5e-4 | 1e-4 |
| Warmup Step | 0.1 | 0.1 | 0.1 |
| Warmup Strategy | Linear | Linear | Linear |
| Image Size | 224 | 224 | 224 |

the input of the different order into the LLM judge using ChatGPT or LLaVA.

## C.4 Implementation of Small LMs

Our ExplainHM model utilizes the T5 encoder-decoder architecture [7, 44] as its foundational framework, specifically utilizing the "flan-t5-base" version. For the extraction of image features, following previous work [41], we adopted the state-of-the-art vision Transformer known as CLIP-ViT-B/32 [42], and this module remains static throughout the training process. To effectively integrate the multimodal information, we incorporated a simple one-head cross-attention mechanism in each layer of the T5 encoder. The maximum length of textual input is set as 512. During the fusion process, the text features are utilized as the query, while the image features act as the key and value. It is noteworthy that these fusion modules were initialized randomly. The dimension $d$ of the hidden states is set as 768, and $d_k$ is set as 384. For the training phase, we provide a comprehensive list of the hyper-parameters in Table 7. Results are averaged over ten random runs. All experiments were conducted using a single V100 32GiB GPU.

## C.5 Prompting GPT-4 for Automatic Evaluation of Explainability

Different from the LLM judge for the detection purpose, as we need to evaluate the quality of the explanations generated from different LLMs like LLaVA and ChatGPT, to avoid the automatic evaluator showing a preference to the side with the same LLM [56], currently the more powerful LLM than LLaVA and ChatGPT, *i.e.*, GPT-4, is the best choice to conduct the explanation evaluation. During the period of this work, the GPT-4 API could be utilized in the language-only modality, similar to ChatGPT, so we extracted the text caption of the meme image by LLaVA as the meme caption to describe the image in the user prompt. For the system prompt to the GPT-4 model, we design the message as follows:

"*Suppose you have been specially designed to perform an explanation evaluation for the harmful meme detection task, you are required to score the provided explanations given the meme text and image. The image and the textual content in the meme are often uncorrelated, but its overall semantic information is presented holistically. Thus it is important to note that you are prohibited from relying on your own imagination, as your goal is to provide the most accurate and reliable score possible.*".

**Figure 5: Prompting LLMs from the harmful argument in the multimodal debate stage, regarding the potential harmfulness label as part of the observed attributes of the textual prompt.**

Moreover, to prompt GPT-4 for the automatic explanation evaluation in each criterion, we designed the template for the user prompt as:

"*Given a Text: [**Meme_text**], which is embedded in an Image: [**Meme_caption**], with a harmfulness label 'harmful', please assign the three explanations respectively with three corresponding score values in Integer, on a rating scale from 1 (worst) to 5 (best) with respect to the [**Criterion**]: 1) [**Explanation_chatgpt**]; 2) [**Explanation_llava**]; 3) [**Explanation_human**].*".

## D PROMPTING LLAVA WITH HARMFULNESS LABELS

Figure 5 illustrates the details of prompting LLaVA from the harmful argument in the multimodal debate stage. We conduct abductive reasoning with LLMs to extract harmfulness rationales (lilac) by the prompt consisting of the meme text (brown), and the label (blue), together with the meme image. Additionally, during our detailed examination of data samples, we discovered a few instances of contentious annotations. More specifically, in the rare instances of incorrect or disputable annotations, even though we engage in abductive reasoning with the incorrect golden label as part of the conflicting rationales in the multimodal debate phase, the content generated by LLMs fails to clearly articulate the reasoning, behind categorizing the meme as the gold truth. It often tends to produce vague statements like "the meme seems to be promoting a harmful and potentially harmful message" without providing a well-structured logical reasoning chain leading to the conclusion. However, on a brighter note, this phenomenon underscores the LLMs' deficiency in logical reasoning knowledge when faced with incorrect annotations. This insight also contributes to a better understanding of our approach's robustness, particularly when dealing with inaccurately annotated data.

## E EFFECT OF RATIONALE ORDER

We can see from Table 8 that the fixed order of the harmful and harmless rationales has not much effect on the harmful meme detection performance of the Small LM judge, under the setting without integration with the LLM judge.

## F PERFORMANCE OF DIFFERENT LM SIZES

To assess the extent to which our approach benefits different versions of the backbone in the Small LM judge, we modify the underlying language models to various variants of different sizes. This allows us to test the generalizability of our approach across different configurations and evaluate its effectiveness in diverse settings, as shown in Table 9.

## G AUTOMATIC EVALUATION OF EXPLAINABILITY ON HARM-C/P DATA

We further provide the results of automatic GPT-4 evaluation on Harm-C and Harm-P data, as shown in Table 10. Note that as there are no existing human-written explanations for the Harm-C and Harm-P data, we only evaluate the text quality of the explanations generated by our model variants from LLaVA and ChatGPT. Although previous work [15] has presented human-written explanations[6], it is labor-intensive and limited that only focuses on the FHM dataset and explains why the meme is harmful or not but without the reasoning thought chains for how the two multimodalities of memes interact with each other to derive the harmfulness. Moreover, the explanations automatically generated from LLMs could provide new benchmarks for future studies about explainable harmful meme detection and automatic evaluation of the explanation quality.

## H HELPFULNESS OF CONFLICTING RATIONALES ON HUMAN SUBJECTS

We have evaluated the detection performance and the text quality of the output explanations, respectively, in the main paper. We further design a human subject study to evaluate the helpfulness of the conflicting rationales for human beings to make correct harmfulness predictions. Specifically, we first randomly selected 100 samples (50 harmful samples and 50 harmless samples) from Harm-C, Harm-P and FHM datasets. Then ten English-speaking evaluators are asked to test on the selected samples. Their average detection performance was 58.50% accuracy. Afterward, we provide the same samples with conflicting rationales from both harmless and harmful arguments for each sample. The average detection performance of the evaluators improved to 77.52% accuracy. The study shows that, by considering both the positive and negative

---

[6]https://github.com/Social-AI-Studio/HatRed

**Table 8: Effect of the rationale order in the input sequence of the Small LM judge.**

| Dataset | Harm-C | | Harm-P | | FHM | |
|---|---|---|---|---|---|---|
| Model | Accuracy | Macro-$F_1$ | Accuracy | Macro-$F_1$ | Accuracy | Macro-$F_1$ |
| Harmless&Harmful | 85.59 | 84.80 | 88.44 | 88.43 | 71.40 | 70.94 |
| Harmful&Harmless | 85.03 | 84.51 | 90.17 | 90.17 | 71.00 | 70.00 |

**Table 9: Detection Performance with different backbone sizes of the Small LM judge.**

| Dataset | Harm-C | | Harm-P | | FHM | |
|---|---|---|---|---|---|---|
| Model | Accuracy | Macro-$F_1$ | Accuracy | Macro-$F_1$ | Accuracy | Macro-$F_1$ |
| Small | 88.42 | 87.44 | 90.08 | 90.07 | 71.40 | 71.07 |
| Base | 87.00 | 86.41 | 90.73 | 90.72 | 75.60 | 75.39 |
| Large | 86.44 | 84.99 | 91.56 | 91.56 | 75.80 | 75.15 |

**Table 10: Automatic GPT-4 evaluation of the explanation quality on Harm-C/P test sets, where the explanations are generated by LLaVA and ChatGPT.**

| Data | Harm-C | | Harm-P | |
|---|---|---|---|---|
| Explanations | LLaVA | ChatGPT | LLaVA | ChatGPT |
| Informativeness | 3.70 | 4.64 | 3.97 | 4.74 |
| Readability | 4.29 | 4.96 | 4.56 | 4.99 |
| Soundness | 3.71 | 4.83 | 3.92 | 4.83 |
| Conciseness | 3.70 | 3.38 | 3.91 | 3.23 |
| Persuasiveness | 3.63 | 4.71 | 3.83 | 4.69 |

aspects of harmfulness, the conflicting rationales can provide human users or checkers with dialectical thinking that allows them to better decode the underlying meaning of memes and mitigate the harmful information.

## I  ERROR ANALYSIS

To better understand the behavior of our harmful meme detection model and facilitate future studies, we conduct an error analysis on the wrongly predicted memes by our proposed explainable framework. To make the analysis more intuitive, we provide each example with both the output explanation for the wrong prediction and the alternative explanation for the ground truth, as shown in Figure 6.

We can analyze that: 1) For the first example in Figure 6(a), we can find that the model was not aware that the "potato" in the meme text is a reference to the people with intellectual disabilities in the image; the same phenomenon is observed in the second example of Figure 6(b), where the females in the meme image are dehumanized as lesser humans that are useful for making sandwiches and ironing clothes related to the words like "sandwich maker" and "ironing board" in the text. The phenomenon shows that, for a few specific vulnerable target types like disabilities and women, there is still room for improvement in understanding how the seemingly uncorrelated texts and images of memes interact;

2) We also presented the third example in Figure 6(c) to investigate the impact of visual artifacts like image quality, occlusion, obscurity, etc. In the example, the girl dressed in red is in obscurity on her face, and the image is also low-quality, which leads to the wrong recognition of "a man in a red hat" in the output explanation. From the alternative explanation for the gold truth, we can also find that the implicit relationships between the rulers and the people of the country, which the occlusion of the Trump avatar and the American flag is intended to express, were not captured. This causes the explanation from the harmful argument to be relatively far-fetched. A similar implication could be observed in the fourth error example in Figure 6(d), which demonstrates that the harmful meme target like race in the image is not actively recognized to build an association with the "cotton". Thus we argue that designing better prompts in the multimodal debate stage, to guide more complete and correct image recognition would further improve the multimodal reasoning between the two modalities of memes;

3) The fifth example in Figure 6(e) illustrates a drawback commonly found in most models designed to detect harmful content - they can be biased. During the training phase, if the model is exposed to harmful content targeting race or religion, it may develop a tendency to label any content related to Muslims as harmful. To address this issue, it may need to employ debiasing techniques in the future.

## J  MORE EXAMPLES OF OUTPUT EXPLANATIONS

We further summarize the following interesting observations about the utility of LLMs' rationales on harmful meme target subcategories: 1) The LLMs could reason well for the offensiveness or hatefulness about the religion, migrant, and LGBT community, since the correlations of the text and images of the aforementioned target subcategories are easy to reason with commonsense knowledge in LLMs. 2) It is slightly poor for LLMs to generate rationales about racism, because there could be some mistakes in recognizing the skin color in the images, resulting from the image color being black and white in a few data examples. 3) In the case of correctly recognizing image features, LLMs can have overall excellent results

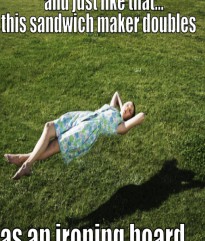

**Meme text:** our dad is a potato farmer

**Output explanation for the wrong prediction (harmless):**
The meme with the text "our dad is a potato farmer" and two people laughing in the image can be considered harmless because it is a lighthearted and playful depiction. It does not intend to mock or belittle individuals with intellectual or physical disabilities. Instead, it uses humor to portray a connection between the two people, likely siblings, and their father's occupation as a potato farmer. The context of the meme is based on a shared joke or bonding moment between the individuals, which is not harmful. It is essential to consider the intent behind the meme and the context in which it is being shared, rather than solely focusing on the text or the image itself.

**Alternative explanation for the gold truth (harmful):**
The meme featuring the text 'our dad is a potato farmer' and two people laughing in the image can be considered harmful as it plays on stereotypes and mocks people with intellectual or developmental disabilities, including Down syndrome. The implication is that the father's disability is something to laugh about or is less dignified than other occupations. This kind of humor perpetuates negative stereotypes and can be hurtful to the individuals and their families who may be affected by such disabilities. It is essential to promote empathy, understanding, and inclusivity by treating all individuals with respect and dignity, regardless of their abilities or disabilities.

(a)

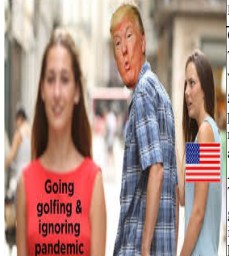

**Meme text:** and just like that... this sandwich maker doubles as an ironing board

**Output explanation for the wrong prediction (harmless):**
The meme is a lighthearted and humorous take on the concept of multitasking, as it shows a woman lying down on a sandwich maker, which is also doubling as an ironing board. The meme implies that the woman is not only enjoying her sandwich but also using the opportunity to get some ironing done simultaneously. The phrase "and just like that... this sandwich maker doubles as an ironing board" highlights the idea of repurposing everyday objects for different tasks. The meme is considered harmless because it is meant to entertain and make people laugh, rather than intending to offend or harm anyone. The contextual background and commonsense knowledge of the meme's viewers allow them to understand the humor in the situation and not take it seriously.

**Alternative explanation for the gold truth (harmful):**
The meme in the image is considered harmful because it makes light of a serious issue related to domestic violence. The woman in the picture is depicted as lying down on a sandwich maker, which is being used as an ironing board, while the text suggests that it is a sandwich maker doubling as an ironing board. The meme trivializes the importance of providing a safe and supportive environment for victims of domestic abuse. Ironically, the meme uses an everyday household appliance like a sandwich maker to make light of a serious situation, which can be interpreted as insensitive and inappropriate. It is essential to be mindful of the content we share and how it might be perceived by others, especially when discussing sensitive topics like domestic violence.

(b)

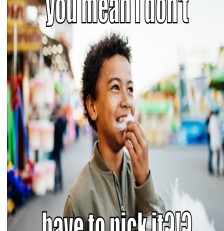

**Meme text:** Going golfing & ignoring pandemic

**Output explanation for the wrong prediction (harmless):**
The meme in the image is a distracted-boyfriend meme, which features a man in a red hat (possibly resembling President Trump) looking at a girl who is being ignored while she is talking about a golf game. The text within the meme implies that the boyfriend is more interested in watching a golf game on TV than paying attention to his girlfriend. The harmfulness of this meme is considered 'harmless' because it is a light-hearted and comical portrayal of a common scenario in relationships, where one person becomes engrossed in an activity or interest while the other person seeks attention. It does not intend to cause offense or harm to the individuals portrayed in the meme or their respective groups. The meme relies on humor and exaggeration to entertain and communicate its message, which is to highlight the boyfriend's lack of attention to his girlfriend.

**Alternative explanation for the gold truth (harmful):**
The meme in the image is considered harmful because it makes light of a serious situation, specifically the COVID-19 pandemic. The text within the meme, "Going golfing & ignoring pandemic", implies that the people depicted in the image are not taking the pandemic seriously and are engaging in leisure activities, such as golfing, instead of focusing on the health crisis. This kind of humor can be seen as insensitive and disrespectful to those who have been affected by the pandemic or those who have lost loved ones. It is important to be mindful of the impact that memes and social media content can have on public perception and the emotional well-being of individuals touched by such serious issues.

(c)

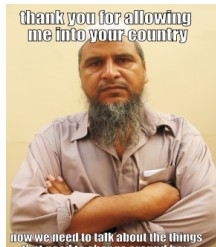

**Meme text:** you mean i don't have to pick it?!?

**Output explanation for the wrong prediction (harmless):**
The meme is a playful and lighthearted depiction of a young boy eating cotton candy, with the text "you mean i don't have to pick it?!?" embedded in the image. This text is likely referencing the common expectation that one must pay for and actively participate in the selection process when purchasing cotton candy at fairs or amusement parks. The meme suggests that the boy is pleasantly surprised to discover that he doesn't have to pick it, implying that he is enjoying a carefree and hassle-free experience. The harmfulness of this meme is reasoned as 'harmless' because it is a light-hearted and comical portrayal, rather than containing any offensive or harmful content. It does not target any specific individual or group, and its humor is derived from a relatable and playful scenario. The meme is likely meant to bring a smile to the viewer's face, rather than causing any distress or harm.

**Alternative explanation for the gold truth (harmful):**
The meme in the image is a playful and lighthearted depiction of a young boy with cotton candy on his face, seemingly surprised or excited about it. The text accompanying the meme says, "you mean i don't have to pick it?!?", which adds to the humor and sarcasm of the scene. The harmfulness of the meme cannot be reasoned as 'harmful' as it is a harmless, fun, and comical representation. It is not intended to cause offense or distress, but rather to entertain and bring a smile to the viewer's face. The meme relies on the viewer's understanding of the playful context and the boy's apparent enjoyment of the cotton candy to create a humorous effect.

(d)

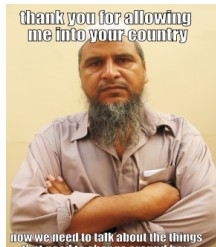

**Meme text:** thank you for allowing me into your country now we need to talk about the things that need to change around here to accommodate my beliefs

**Output explanation for the wrong prediction (harmful):**
The meme is considered harmful because it mocks and belittles the idea of cultural sensitivity and understanding. By using a picture of a bearded man with an ethnic appearance and incorporating it with a text that appears to make light of the situation, the meme perpetuates stereotypes and promotes intolerance. This kind of content can be offensive to people from different cultural backgrounds and can contribute to a negative atmosphere by promoting divisiveness and undermining the importance of respectful communication across different communities. It is essential to approach sensitive topics with care and empathy, and sharing such harmful memes only serves to hinder meaningful conversations and understanding.

**Alternative explanation for the gold truth (harmless):**
The meme is a playful and harmless depiction of a man with a beard, possibly of Muslim origin, welcoming someone to his country. It is important to note that the meme is not intended to be taken seriously and is meant to be humorous. The statement "now we need to talk about the things that need to change around here to accommodate my beliefs" is a lighthearted and exaggerated way of expressing the idea of cultural exchange and understanding. The meme is not promoting any harmful or discriminatory views, but rather, it is a playful and satirical commentary on the idea of adapting to a new culture. It is essential to consider the context and intent behind the meme before jumping to conclusions or interpreting it negatively.

(e)

**Figure 6: Examples of wrongly predicted memes by our proposed framework with the ground truth (a) harmful, (b) harmful, (c) harmful, (d) harmful and (e) harmless.**

in rationale generation with our proposed prompting mechanism (Figure 5). Except for those generated from LLaVA, more case examples related to the output explanations generated from ChatGPT can be found in Figure 7.

## K  LIMITATIONS AND FUTURE WORK

There are multiple ways to further improve this work:

- Overall, the explainability of this work focuses on that the model's decision is explainable with the rationales. However, there might be a deeper level of explainability of the model that is not touched on in this paper, which is to explain how a neural model works internally. We would further improve our research to facilitate the interpretability of the model architecture.

- We heuristically designed the prompt of LLMs for the multimodal debate in only one turn. But in some error examples, the generated text may miss the details of the meme like the race. We would further update our prompt for the design of multi-turn debates with LLMs, to activate the commonsense reasoning knowledge related to vulnerable targets in harmful content, improve the visual feature extraction for exploring better multimodal reasoning thoughts, and avoid several common deficiencies of existing language models including hallucination and limited generalization as much as possible.

- Despite this work utilizing GPT-4 for the automatic evaluation of explanation quality, the evaluation results still have minor gaps with the human subject study, like GPT-4 tends to judge the explanations from ChatGPT with higher scores than those from other sources or models. Moreover, if GPT-4 is incorporated into the multimodal debate stage, we need to seek more powerful language models to evaluate the explanations generated by GPT-4. Thus more accurate automatic evaluation of the explanation quality is needed, meanwhile, more comprehensive human subject studies could be conducted on a larger crowd of evaluators in an organized manner.

- Generally, the distribution drift in datasets over time is a potential limitation for almost all data-driven tasks, especially for the memes on the Web. However, one of the contributions of this work is proposing a novel paradigm to leverage commonsense reasoning knowledge in LLMs for the harmful meme detection task. The proposed framework is general enough, which should still work with newly released stronger LLMs or new meme data appearing on the Web. For example, in the future, we could publish a plug-and-play interface to incorporate a broader range of LLMs into our framework for the multimodal debate stage, even the GPT-4V[7] if there is sufficient financial support for some users.

- Although harmfulness is defined much broader than hatefulness or offensiveness in previous literature [40], we have evaluated and analyzed our approach on the meme dataset related to hate speech [22]. In the future, we would try to incorporate more of the related meme datasets beyond our task to further broaden the boundaries of this research, such as offensiveness [49], sexism [12], or cyberbullying [53], etc.

## L  ETHICS AND BROADER IMPACT

The purpose of this work is to prevent the spread of harmful meme information and to ensure that people are not subjected to prejudice or racial and gender discrimination. Nevertheless, we are aware of the potential for malicious users to reverse-engineer and create memes that go undetected or misunderstood by ExplainHM-trained AI systems. This is strongly discouraged and condemned. Intervention with human moderation would be required in order to ensure that this does not occur. Research indicates that evaluating harmful or hateful content can have negative effects. To protect our human evaluators, we establish three guidelines: 1) ensuring their acknowledgment of viewing potentially harmful content, 2) limiting weekly evaluations and encouraging a lighter daily workload, and 3) advising them to stop if they feel overwhelmed. Finally, we regularly check in with evaluators to ensure their well-being. Another consideration is the usage of Facebook's meme dataset; users will have to agree with Facebook's usage agreement to gain access to the memes. The usage of Facebook's memes in this study is in accordance with its usage agreement. All the datasets only include memes and do not contain any user information.

---

[7]https://openai.com/research/gpt-4v-system-card

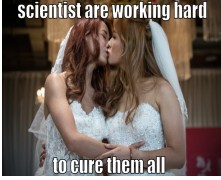

**Meme text:** scientist are working hard to cure them all
**Label:** Harmful

**Output explanation:**
The harmfulness of this meme is reasoned as harmful due to the potential negative impact it may have on certain individuals or communities. the image depicts two women in wedding dresses sharing an intimate moment, possibly celebrating their recent marriage. the text states, "scientists are working hard to cure them all." this combination of image and text can be interpreted as mocking or belittling same-sex relationships, suggesting that being gay or lesbian is something that needs to be "cured." this perpetuates harmful stereotypes and discrimination against the lgbtq+ community. by associating the image of a loving same-sex couple with the idea of needing a cure, the meme promotes a harmful and offensive message. it is important to recognize and reject such harmful content to foster inclusivity and respect for all individuals.

(a)

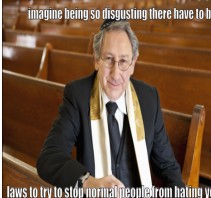

**Meme text:** imagine being so disgusting there have to be laws to try to stop normal people from hating you
**Label:** Harmful

**Output explanation:**
The harmfulness of this meme can be reasoned based on the derogatory and offensive nature of the embedded text. the text implies a strong negative sentiment towards a specific group of people, suggesting that they are disgusting and deserving of hatred. this kind of language promotes discrimination, prejudice, and animosity towards individuals based on their religious or ethnic background. the image itself adds to the harmfulness by depicting a man wearing religious attire associated with the jewish faith in a church setting. this combination of elements can be seen as mocking or disrespectful towards both the jewish and christian communities. it may perpetuate stereotypes, fuel religious tensions, and contribute to the spread of hate speech. overall, the text and image together create a harmful message that promotes hatred and intolerance towards a particular religious group. it is important to recognize and address such harmful content to foster a more inclusive and respectful online environment.

(b)

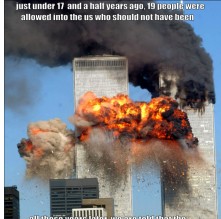

**Meme text:** just under 17 and a half years ago, 19 people were allowed into the us who should not have been all these years later, we are told that the answer to safety is to allow everyone in
**Label:** Harmful

**Output explanation:**
The harmfulness of this meme can be reasoned as harmful due to the combination of the text and the image. the text mentions an event that occurred "just under 17 and a half years ago" where "19 people were allowed into the us who should not have been." it implies that this event has had long-lasting consequences. the image accompanying the text depicts the twin towers of the world trade center collapsing due to a massive explosion, with flames and a huge fireball. the text and image together evoke the tragic events of 9/11, a terrorist attack that resulted in the loss of thousands of lives. the mention of people being allowed into the us who should not have been suggests a critique of immigration policies. the meme implies that allowing everyone into the country, as a response to safety concerns, is a flawed approach. the harmfulness of this meme lies in its potential to exploit a tragic event for political or ideological purposes. it may evoke strong emotions and perpetuate negative stereotypes or biases towards certain groups of people. additionally, it can be distressing and triggering for individuals who have personal connections to the 9/11 attacks. overall, the combination of the text referencing immigration and the image depicting the 9/11 attacks creates a harmful and potentially divisive message.

(c)

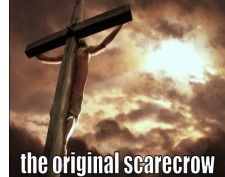

**Meme text:** the original scarecrow
**Label:** Harmful

**Output explanation:**
The harmfulness of this meme can be reasoned based on the combination of the text and image. the text, "the original scarecrow," does not directly relate to the image of jesus christ on the cross. however, when considering the overall semantics, the image depicts a graphic and intense scene of the crucifixion, emphasizing the suffering and intense emotions experienced by jesus during his crucifixion. the use of the term "scarecrow" in the text could be seen as disrespectful or mocking towards the religious significance of jesus' crucifixion. this could potentially offend and upset individuals who hold religious beliefs related to jesus' sacrifice and the crucifixion. furthermore, the graphic nature of the image, with a dark and dramatic sky in the background, may contribute to the harmfulness of the meme. it could be perceived as insensitive or offensive, as it portrays a sacred and deeply meaningful event in a potentially disturbing or sensationalized manner. considering these factors, the combination of the text and image in this meme could be seen as harmful due to its potential to offend religious sensibilities and its graphic depiction of a sacred event.

(d)

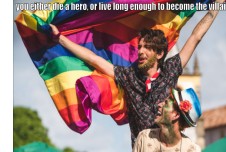

**Meme text:** you either die a hero, or live long enough to become the villain
**Label:** Harmless

**Output explanation:**
The meme is classified as harmless because it combines a popular quote with an image depicting a celebration of diversity and inclusivity. the text, "you either die a hero, or live long enough to become the villain," is a well-known line from the movie "the dark knight." in the context of the meme, it suggests that individuals have the potential to change over time. the image shows two men holding a colorful rainbow flag, which is commonly associated with lgbtq+ pride. the flag represents the diverse identities and experiences within the lgbtq+ community. the festive atmosphere and the presence of a hat indicate a celebration or event related to lgbtq+ pride. by combining the quote with the image, the meme conveys a positive message about embracing one's true self and the journey of personal growth. it promotes acceptance, inclusivity, and the celebration of diversity. there is no harmful or offensive content present in the meme, making it harmless overall.

(e)

**Figure 7: Examples of output explanations generated from ChatGPT.**

