# OpenReview forum: "Towards Explainable Harmful Meme Detection through Multimodal Debate between Large Language Models"
_ACM.org/TheWebConf/2024/Conference — TheWebConf24 Oral_

### Official Review · Reviewer_J8wT · 2023-11-23

**Novelty:** 3
**Technical Quality:** 3

**Review:**

This paper aims to enhance harmful meme detection using powerful multimodal LLMs such as LLaVA and GPT-4 to generate dialectical reasoning texts. The conducted experiments provide evidence of the effectiveness of the proposed methods.

Weakness:
1. While Figure 3 visualizes harmful arguments from LLMs to support correct prediction samples, it should also include visualizations of harmless arguments for comparison. Additionally, it would be beneficial to provide examples of incorrect samples where both harmful and harmless arguments in the debate are presented.
2. The notations used in the contexts and tables should be consistent.
3. I am curious about the dialectical argument in LLMs depicted in Figure 1. Are they the same as the ones analyzed in the introduction?

**Questions:**

See Weakness.

**Reviewer Confidence:**

3: The reviewer is confident but not certain that the evaluation is correct

**Scope:**

4: The work is relevant to the Web and to the track, and is of broad interest to the community

---

### Official Review · Reviewer_AfTK · 2023-11-24

**Novelty:** 5
**Technical Quality:** 4

**Review:**

This paper proposes an innovative approach to detect harmful memes and provide readable explanations for our detection decisions. The task presents a significant challenge due to the implicit meaning embedded in memes, which is not explicitly conveyed through the surface text and image. Existing harmful meme detection methods do not present readable explanations that unveil such implicit meaning to support their detection decisions.  The proposed approach uses Large Language Models (LLMs) to generate multimodal debate between conflicting rationales from both harmless and harmful positions. The authors then fine-tune a small language model as the debate judge for harmfulness inference, to facilitate multimodal fusion between the harmfulness rationales and the intrinsic multimodal information within memes.

Pros:

The novelty of this work lies in its explainable approach to detect harmful memes, achieved through reasoning over conflicting rationales from both harmless and harmful positions. The proposed approach achieves much better performance than state-of-the-art methods and exhibits a superior capacity for explaining the meme harmfulness of the model predictions. The authors also conduct extensive experiments on three public meme datasets to demonstrate the effectiveness of their approach.

Cons:

Besides leveraging LLM, the algorithmic contribution of this work is slim.

**Questions:**

N.A.

**Ethics Review Description:**

N.A.

**Reviewer Confidence:**

3: The reviewer is confident but not certain that the evaluation is correct

**Scope:**

3: The work is somewhat relevant to the Web and to the track, and is of narrow interest to a sub-community

---

### Official Review · Reviewer_91Qv · 2023-11-25

**Novelty:** 6
**Technical Quality:** 6

**Review:**

The paper discusses the rise of memes as multimodal entities combining images and text, often used humorously but also capable of causing harm by fostering political and socio-cultural divisions, as seen in examples like anti-vaccination memes during the COVID-19 pandemic.

It highlights the challenges in detecting harmful memes, which require understanding their implicit meanings beyond superficial analysis, and proposes an approach using Large Language Models (LLMs) for more accurate and explainable harmful meme detection.

The study introduces an explainable approach, ExplainHM, which uses a multimodal debate between LLMs to generate rationales from both harmless and harmful perspectives, improving the detection of harmful memes and providing informative explanations for their harmfulness.


overall an interesting and important study.
well-written with space for improvement especially with respect to the presentation of results.

for instance, table 5 what is the range and meaning of the values? likert scale i suppose but what is high and low informative is not clear. Also, the deviation of the algorithmic to human evaluation is quite large in several cases.

limitation and implications are fundamental for this study and are only mentioned in the end of the appendix.

The error analysis is also an important part of this study.
I would recommend that the authors reconsider and reorganise the manuscript including the limitations and ethical implications in the main text.

**Questions:**

for instance, table 5 what is the range and meaning of the values? likert scale i suppose but what is high and low informative is not clear. Also, the deviation of the algorithmic to human evaluation is quite large in several cases.

limitation and implications are fundamental for this study and are only mentioned in the end of the appendix.

The error analysis is also an important part of this study.
I would recommend that the authors reconsider and reorganise the manuscript including the limitations and ethical implications in the main text.

**Reviewer Confidence:**

4: The reviewer is certain that the evaluation is correct and very familiar with the relevant literature

**Scope:**

4: The work is relevant to the Web and to the track, and is of broad interest to the community

---

### Official Review · Reviewer_zVRd · 2023-11-26

**Novelty:** 6
**Technical Quality:** 6

**Review:**

In this work, authors propose a novel method for the task of harmful memes detection. This method is based on the fascinating concept of multimodal debate between LLMs: two LLMs debate on whether the given example is in one class or the other, and then an ad-hoc "judge" network decide the output based on their input.

### Strong points

S1. The proposed method of multimodal debate between LLMs seem novel, interesting, and fruitful. It is very well justified in Introduction by contextualizing it within the idea of consensus formation through debate.

S2. The experimental results are convincing: there is an ablation analysis that shows how the complexity of the method is necessary to reach optimal perform, and also the proposed method seems to offer an advantage with respect to state of the art.

S3. The proposed method is also showcasing an interesting concept of classification methods that are able to provide textual explaination as a by-product.

### Weak points

W1. As all studies based on ChatGPT, this work suffers from

- (a) Reproducibility: the results obtained with this method are subject to undetectable changes made to ChatGPT by OpenAI. No researcher can be sure if the results they obtain are the same as the authors'.

- (b) Black box explainations: it is impossible for other researcher to further investigate the quality of the explainations provided by ChatGPT, e.g. to understand if they are the result of some specific item in the training set.

- (c) Test set leakage: this work employs a standard data set to evaluate the performance of the proposed approach. How can authors be sure that a part of this data set (or some social media discussion about it, or a textual representation of it) was not used for training ChatGPT?

- (d) Generalizability: do the authors believe that they have discovered a general technique that works with LLMs, or a specific feature of a product sold by OpenAI?

W2. The introduction of the task is often very simplistic, especially in the definition of 'memes' and 'harmful'. In fact authors starts by defining "a meme consists of a picture combined or embedded with a concise textual component" but the definition of meme is often very different from this one (starting from the original definition by Richard Dawkins) and often they are defined based on their ability to be modified and spread [1, 2]. Similarly, in the next lines they define a meme being harmful by the fact that it creates political divisions, implying that any political division is a harm (including, for example, the division created by anti-apartheid activists in U.S. in the '60s). This concept feels very authoritarian in nature.


[1] Eline Zenner and Dirk Geeraerts. 2018. One does not simply process memes: Image macros as multimodal constructions. Cultures and traditions of
wordplay and wordplay research (2018), 167–194.

[2] Stephen D Reese, Jr Gandy, and August E Grant. 2001. Prologue—Framing public life: A bridging model for media research. In Framing public life.
Routledge, 23–48.

Edit:

I have carefully read the authors' response and I am sorry for the misunderstanding: while Section 3.2 is indeed quite clear, the presence of ChatGPT in all experiments, together with the sentence "LLaVA and ChatGPT are selected as the representative LLMs from the vision and language perspectives", made me think that ChatGPT was used as an integral part of the proposed framework, working side by side with LLaVa for the language part. I see now that instead is being used only as a text-only baseline in the ablation studies. My suggestion is to make this more clear and motivated from the beginning of section 4.3 – even if I recognize it was a slip on my part. Anyway, I thank the authors for the careful answer to all of my comments; I will update my score to reflect an increase in technical quality since the issue of reproducibility is seriously taken into consideration.

At the same time, I believe that the answer to W2 is still unsatisfactory: while not giving formal definitions, the introduction of the object of study should be more precise.

**Questions:**

1. How can authors be sure that a part of this data set (or some social media discussion about it, or a textual representation of it) was not used for training ChatGPT?
2. Do authors believe that their work is about a specific feature of this OpenAI product or is it a general property of LLMs?
3. Is this work reproducible?
4. Could the same study be done using an openly available LLM instead of ChatGPT?

**Ethics Review Description:**

-

**Reviewer Confidence:**

3: The reviewer is confident but not certain that the evaluation is correct

**Scope:**

4: The work is relevant to the Web and to the track, and is of broad interest to the community

---

### Decision · Program_Chairs · 2024-01-22

**Decision:**

Accept (Oral)

**Comment:**

In this work, the authors propose an explainable approach for detecting harmful memes, utilizing Large Language Models (LLMs) to generate multimodal debates and fine-tuning a small language model as a judge, achieving superior performance and explanation capacity in identifying implicit harmful patterns in memes. Reviewers generally agree that this is important, timely, novel, and technically interesting. Furthermore, this paper should be of interest to the conference.

 I encourage the authors to include the limitations section as part of the main text in their camera-ready submission. As the reviewers rightly point out, this is crucial for contextualizing the findings of the apper.